# Evaluation of Latent Models Assessing Physical Fitness and the Healthy Eating Index in Community Studies: Time-, Sex-, and Diabetes-Status Invariance

**DOI:** 10.3390/nu13124258

**Published:** 2021-11-26

**Authors:** Scott B. Maitland, Paula Brauer, David M. Mutch, Dawna Royall, Doug Klein, Angelo Tremblay, Caroline Rheaume, Rupinder Dhaliwal, Khursheed Jeejeebhoy

**Affiliations:** 1Department of Family Relations & Applied Nutrition, University of Guelph, 50 Stone Road E, Guelph, ON N1G 2W1, Canada; pbrauer@uoguelph.ca (P.B.); phcnutr@uoguelph.ca (D.R.); 2Department of Human and Health Nutritional Sciences, University of Guelph, Guelph, ON N1G 2W1, Canada; dmutch@uoguelph.ca; 3Department of Family Medicine, University of Alberta, Edmonton, AB T6G 1C9, Canada; doug.klein@ualberta.ca; 4Department of Kinesiology, Faculty of Medicine, Université Laval, Quebec City, QC G1V 0A6, Canada; angelo.tremblay@kin.ulaval.ca; 5Centre de recherche Nutrition, Santé et Société (NUTRISS), INAF, Quebec City, QC G1V 0A6, Canada; 6Department of Family Medicine and Emergency Medicine, Faculty of Medicine, Université Laval, Quebec City, QC G1V 0A6, Canada; Caroline.Rheaume@fmed.ulaval.ca; 7Canadian Nutrition Society, Ottawa, ON K1C 6A8, Canada; rupinder@cns-scn.ca; 8Departments of Nutritional Sciences and Physiology, University of Toronto, Toronto, ON M5S 1A8, Canada; khushjeejeebhoy@hotmail.com

**Keywords:** physical fitness, diet quality, factor analysis, structural equation modeling, measurement equivalence/invariance, metabolic syndrome, cardiometabolic health

## Abstract

Accurate measurement requires assessment of measurement equivalence/invariance (ME/I) to demonstrate that the tests/measurements perform equally well and measure the same underlying constructs across groups and over time. Using structural equation modeling, the measurement properties (stability and responsiveness) of intervention measures used in a study of metabolic syndrome (MetS) treatment in primary care offices, were assessed. The primary study (N = 293; mean age = 59 years) had achieved 19% reversal of MetS overall; yet neither diet quality nor aerobic capacity were correlated with declines in cardiovascular disease risk. Factor analytic methods were used to develop measurement models and factorial invariance were tested across three time points (baseline, 3-month, 12-month), sex (male/female), and diabetes status for the Canadian Healthy Eating Index (2005 HEI-C) and several fitness measures combined (percentile VO_2_ max from submaximal exercise, treadmill speed, curl-ups, push-ups). The model fit for the original HEI-C was poor and could account for the lack of associations in the primary study. A reduced HEI-C and a 4-item fitness model demonstrated excellent model fit and measurement equivalence across time, sex, and diabetes status. Increased use of factor analytic methods increases measurement precision, controls error, and improves ability to link interventions to expected clinical outcomes.

## 1. Introduction 

### 1.1. Lifestyle Treatment of Cardio-Metabolic Conditions 

Significant progress has been made in demonstrating the overall benefits of personalized lifestyle counselling in prevention of cardiometabolic conditions. Cardiometabolic risk (CMR) conditions include various combinations of prediabetes and type 2 diabetes, hypertension, dyslipidemia and higher visceral abdominal fat accumulation, as typically assessed by waist circumference. Several large clinical trials have demonstrated reductions in cardiovascular (CVD) mortality and diabetes incidence, namely the PREDIMED study [1,2] and the Diabetes Prevention Program [3,4,5] and subsequent studies [6,7]. CMR conditions and diseases are a major and growing health burden in many countries, as obesity continues to increase worldwide [8]. Excess body weight is associated with adverse metabolic effects in a sizable minority and become more prominent in middle age. These adverse effects manifest as the already mentioned conditions, as well as the combination described as metabolic syndrome (MetS). MetS is defined as three or more indicators, including higher waist circumference, higher blood pressure, dyslipidemia characterized by low high-density lipoprotein and elevated triglyceride levels, and elevated glucose levels [9]. The various clinical definitions describe overlapping populations [10], and different combinations of risk factors likely differentially affect CVD risk [11]. For example, people with MetS have approximately double the CVD risk as people without MetS [12].

Worldwide prevalence of some of the risk factors like hypertension, obesity and type 2 diabetes are well documented [13], while prediabetes [14] and MetS have less often been assessed in national surveys [15]. In Canada, 21% of adults 20–79 years old had MetS in the 2012–2013 survey [16], whereas in the United States (US) 33% of adults aged 20 and older met the criteria for the condition in NHANES 2002–2013 [17]. Among people 60 years and older, 39% of Canadians and 46% of Americans from the same analyses had MetS [16,17]. Ongoing costs of CMR are substantial, as confirmed in a 2016 US study of the Medical Expenditure Panel Survey. Among those with three or four risk factors (mostly MetS) compared to those with none of the CMR conditions, health care utilization was 50% higher, days missed from work 75% higher and yearly health care costs more than twice as high [18]. In addition, recent experience with COVID-19 has confirmed increased risk of severe disease in the presence of these CMR conditions, although estimates of excess risk vary [19,20].

### 1.2. Measurement Issues

All relevant practice guidelines for CMR conditions promote lifestyle change in a general way [21,22,23,24]. Effective lifestyle services to treat CMR are not, however, routinely being offered within health care in Canada [25] and elsewhere, and multiple issues are involved, including structural issues like lack of resources and expertise in family medicine practices, and clinician perceptions of poor effectiveness of lifestyle programs in practice (the efficacy-effectiveness gap) [6]. Focusing on the efficacy-effectiveness gap, key challenges for researchers include: (1) measurement challenges in assessing diet and exercise in typical community and healthcare settings, (2) measurement issues in identifying the key aspects of the intervention processes, and (3) linking process indicators to key changes in clinical measures at the individual level.

Better measures of diet and exercise (issue 1) that are relatively simple to collect, reliable and could validly assess the achieved level of diet and fitness status at each time point and over time are a priority. With a focus on improving measures, it may be possible to identify key aspects of interventions and better link them to clinical changes.

Assessment of the measurement properties of diet quality and physical fitness measures is the focus of this secondary analysis of a primary care-based lifestyle study [26]. A 19% reversal of MetS and reduction in CVD risk score, as measured by PROCAM risk score (analogous to the Framingham risk score but for MetS) [27] was seen in this one-year study overall, as previously published [26]. Unpublished data showed changes in individual scores for diet quality and aerobic capacity were not correlated with individual changes in PROCAM scores as was hoped, given the overall group results. Only changes in waist circumference were correlated with PROCAM score. Therefore, we asked if measurement error in the intervention measures could have accounted for the lack of associations, and secondly whether changes in one behavior could have had effects on the other. This latter question came from consideration of the potential for a carry -over effect, as discussed in the multiple behavior change literature [28]. Diet and physical activity interventions are the most commonly studied multiple risk behavior interventions [29]. To explore these possibilities, a detailed structural equation modeling (SEM) analysis was undertaken.

First, the physical activity and diet measures are briefly reviewed, followed by an introduction to SEM, which represents a melding of factor analysis and path analysis into one comprehensive statistical methodology. In general, a structural equation model consists of two parts: (1) the measurement model, which links observed variables to latent variables via a confirmatory factor analysis, and (2) the structural model linking latent variables to each other via systems of simultaneous equations [30]. This measurement analysis uses both exploratory factor analysis (EFA) and confirmatory factor analysis (CFA) with detailed explanations of the modeling process. The analysis for carry-over effects was conducted with the resulting latent variables.

### 1.3. Fitness Assessment

Fitness assessment in community- or primary care-based studies may include multiple tests to determine measures of some or all of the four main health-related fitness components: cardio-respiratory fitness or aerobic capacity, muscle fitness or strength, flexibility, and body composition, based on a wide range of standardized procedures [31,32]. A key measure of cardiorespiratory fitness employed in many studies is an estimate of maximal oxygen consumption (VO_2_ max), providing an indicator of both cardiac and pulmonary functioning. VO_2_ max may be expressed as an absolute rate in litres of oxygen per minute (L/min) or in terms of percentiles relative to age-and sex-based averages. Accurate VO_2_ max measurement requires physical effort sufficient in duration and intensity to put the aerobic energy (i.e., cardiorespiratory) system through its range of capacity. A treadmill or exercise bike is used to vary exercise intensity progressively while measuring pulmonary function and the chemical composition of inhalation/exhalation air for the oxygen/carbon dioxide ratio. Accurate assessment of VO_2_ max is beyond the capacity of community studies, so many groups have created various sub-maximal exercise-based and non-exercise-based estimation equations for clinical practice [32]. Significant error has been demonstrated using these less accurate methods and research is underway to develop tools that can be used in clinical practice [33]. In the meantime, a variety of approaches have been used.

### 1.4. Diet Quality Assessment

Assessment of diet in community-based intervention studies remains challenging, given the complexity of diet with many foods eaten daily and the large day-to-day variation in intake. Personalized diet counselling or therapy for CMR conditions, including MetS, involves two main approaches [34]; a weight loss focus as exemplified by the Diabetes Prevention Program [3] versus a focus on diet quality, as exemplified by the PREDIMED study, which promoted a Mediterranean diet [2]. Therefore, multiple diet assessment methods were used to take advantage of the complementary strengths and limitations of different tools, specifically recalls coupled with a food frequency questionnaire (FFQ) approach. In North America, the use of the Healthy Eating Index (HEI) was originally developed for epidemiological studies, as it is scored against the benchmark of the US Dietary Guidelines (the basis of nutrition policy in the US), and has had extensive development, and population data are available for comparison [35]. More recently, its use has been reviewed in CMR intervention studies [36]. Other diet quality tools used in intervention studies include various versions of the Mediterranean Diet Score (MDS) [37], as well as scores based on different aspects of diet (see Miller et al. for recent review [38]). Interest in the use of diet quality scores in lifestyle intervention studies has been growing, as they provide a summary measure that can potentially be linked to clinical outcomes. Therefore, this analysis is timely.

### 1.5. Structural Equation Modeling and Measurement Equivalence/Invariance (ME/I)

Accurate measurement and representation of summary indices and measures requires assessment of measurement equivalence/invariance (ME/I) to demonstrate that the items/tests/measurements perform equally well and measure the same underlying constructs across groups and/or over time [39,40,41,42,43]. The longitudinal design of the current study allows for analysis of a combination of cross sectional and change models, as well as assessment of ME/I [41]. This requires examining simultaneous relationships between constructs in the SEM framework. Appropriate baseline sampling allowed us to understand the extent (i.e., prevalence) of cardiovascular risk, profiles of dietary behaviors, and the initial and subsequent levels of physical activity/fitness. The repeated measures data allows each participant to serve as their own baseline or control, as well as measuring changes in each of the two areas of concern (i.e., dietary/nutritional behavior, physical activity/exercise). Information garnered from cross-sectional models and the evaluation of appropriate measurement models developed through EFA and CFA methods will help inform models to be used to predict changes within constructs as well as testing structural relationships between constructs. As noted by Hayduk [39] creation of latent variables relies on accurate measurement of observed constructs.

With appropriate conceptual models established and ME/I assessed, relationships between the latent HEI and physical activity/fitness constructs over time were assessed to see if changes in one lifestyle intervention were associated with changes in the other over time, in line with the emerging area of multiple risk behavior interventions. The basic argument is that experiences, skills, knowledge and self-efficacy can be carried-over to different behaviors and domains [28].

## 2. Methods

### 2.1. Data from Original Study

The data come from a non-randomized 12-month feasibility study for lifestyle treatment of MetS conducted from 2012–2015 at three Canadian primary care clinics in three different provinces (Edmonton, Alberta, Toronto, Ontario, and Quebec City, Quebec) [26]. All participants were recruited by their primary care physicians. Inclusion criteria included: (1) adults at least 18 years age; (2) a body mass index (BMI) less than 35; and (3) presence of at least 3 out of 5 criteria for MetS [9]. Exclusion criteria included relevant medical, safety or logistic reasons, as described in the primary paper [26]. The study plan is shown in Figure 1. Study data were obtained from the patients’ medical charts and entered into a secure online data capture system (Research Electronic Data Capture; REDCap: http://www.projectredcap.org/; accessed on 1 November 2021) by locally designated clinic staff. The current sample was comprised of 293 adults, aged 18–81 years old (mean 59 years), and 52% female.

Physician review with participants occurred quarterly. As noted in Figure 1, lifestyle intervention consists of weekly appointments for 12 weeks, followed by monthly appointments and interactions for up to one year by locally employed Registered Dietitians (RD) and kinesiologists. All interventions were personalized, and practitioners were trained and supported by the research team [44,45]. The patient experience was highly positive, as documented by focus groups and a questionnaire [46]. This analysis uses the baseline, 3-month and 12-month data.

### 2.2. Available Measures

#### 2.2.1. Physical Activity/Fitness

Aerobic fitness of participants was assessed by a methodology described by Ebbeling et al. to estimate maximal oxygen consumption [47], using a submaximal aerobic fitness test that is considered safe and appropriate for low risk, apparently healthy, non-athletic adults 20–59 years of age. A steady state heart rate is established after a warm-up by altering the treadmill speed at a 5% incline for 4 min, as described in detail elsewhere [48]. Both speed (in miles/h) and heart rate (bpm) are required in the calculation of this version of VO_2_ max. The measure was further adjusted to create a percentile score relative to others in the same age-sex category. Other measures of exercise output and fitness assessed muscular strength, flexibility, and endurance [45,48]. Each of these measures interacts with and is dependent upon the fitness level of the cardiopulmonary system. These various measures were not combined into an overall fitness score in the original study [48,49].

#### 2.2.2. Diet Quality—HEI-C

To calculate the Canadian version of the HEI (HEI-C) [50,51], a FFQ was developed to assess the average number of servings of food groups eaten over the past month and then scored according to specific age and sex criteria based on Canada’s Food Guide (CFG) 2007 recommendations and serving sizes [52] (see Appendix A). The scores for the moderation components, energy (kcal) from saturated fat and other foods (as a percentage of total energy), and sodium (in milligrams) were calculated from the results of two 24-h recalls, done about one week apart, at baseline, 3-months and 12-months, as previously described [50]. The dietary intake data were collected by the RDs at each centre and analyzed centrally to maintain quality control using a comprehensive nutrient analysis program (ESHA Food Processor—Canadian Version 10, Salem, OR, USA) and double data entry.

#### 2.2.3. Other Variables

Many other clinical measures were collected in the dataset, including medical diagnoses and medication usage [26]. While all participants met the criteria for MetS, prevalence of specific features vary in different samples [53], and in the primary study, approximately half of the sample had a formal diagnosis of type 2 diabetes mellitus (DM). As this was a main clinical issue, we assessed by DM status.

### 2.3. Analytics Plan

Correlation matrices were examined for all items from the HEI-C, physical activity/fitness measures, and the PROCAM scores to confirm previous findings. Test-retest correlations across time points were also examined.

Baseline data were considered the first step to establishing the measurement models for HEI-C and physical activity/fitness. Upon establishing an acceptable fit for the baseline models, longitudinal extensions of measurement models were examined. Females were utilized for model development, with replication/extension to males. For disease status, models were initially tested on the no-DM group (i.e., MetS but no diagnosis of DM), then extended to examine those with DM. The fit of all models was assessed using model chi square (χ^2^; non-significant result is desirable but often unlikely in larger samples), Comparative Fit Index, (CFI > 0.90), Non-Normed Fit Index (NNFI, also known as the Tucker-Lewis Index NNFI > 0.90), Root Mean Square Error of Approximation (RMSEA range 0.05–0.08 accepted, smaller values indicate better fit) (see Table 1 and Table 2). The general recommendation is to evaluate model fit through consensus of an array of fit indexes. These fit indices are consistent with recommendations from Bentler [54], Cheung and Rensvold [55], Kline [56], Lai and Green [57], McNeish, An and Hancock [58] and others. Statistically significant χ^2^ values are not unusual in larger samples or more complex models, and the model(s) fit may still be acceptable.

Comparisons of the same constructs over time or in different groups assumes the tests/items demonstrate factorial invariance and this has been described as the most important empirical question to address whenever multiple groups or time points are present [59]. Invariance testing involves a series of nested constraints that examine whether the variance/covariance relationships among variables operate similarly across groups/points of measurement. Multiple tests of ME/I were conducted following the strategies described by Hayduk [39], Little [59], Meredith [60], Millsap [61], Vandenberg and Lance [41], and van de Schoot et al. [42].

Configural invariance, weak metric invariance, strong invariance of measurement intercepts, and strict invariance of the uniqueness or error terms were all examined, as indicated in Table 1 and Table 2. Configural invariance is also called pattern invariance, meaning the same variables are loading onto the same factors across groups or over time. A lack of configural invariance in the physical activity/fitness model would be demonstrated if a measure like speed was important at baseline but not later in the study (or if it was salient for men but not for women). Configural invariance says nothing about the magnitude of the factor loadings, simply that the same variables load onto the same factors across groups [43]. Weak metric invariance, also called factor loading invariance, is a test of the equivalence of the magnitude or size of the factor loadings across groups or time (Little [59], Meredith [60], Vandenberg and Lance [41]). In addition to the same variables loading onto the same factors across groups (or over time), the relationship or proportionality of the loadings on the factors is demonstrated to be the same (i.e., the rank order and the size of the factor loadings are consistent across comparisons). Tests of weak metric or factor loading invariance are often the highest level of ME/I accomplished and finding this evidence is a sizable accomplishment in complex models. There is some disagreement about how difficult weak invariance is to obtain as Horn, McArdle and Mason [62] suggested that configural invariance is often the best one can hope to obtain in social science data. Physiological measure or laboratory values often demonstrate more precision than self-reported data. Little [59] provided a different perspective, suggesting weak invariance is often attained, whereas invariance of intercepts, is much more important and difficult. Every observed variable/item in the SEM model has an intercept and tests of invariance of these intercepts is important if one hopes to examine mean comparisons between groups or over time (as is implicitly done in analysis of variance models). This is often referred to as strong invariance (also known as scalar invariance or intercept invariance) [59]. Finally, the most restrictive form of invariance, and one that is often impossible to attain, is strict invariance (i.e., demonstrating the equality of the error/residual/uniqueness terms across groups/time—also known as error variance invariance or residual invariance). Little [59] suggests this level is overly restrictive and argues even if found, strict invariance does not ensure a “better” level of invariance. Therefore, testing for strict invariance was undertaken, but models were not rejected based on a lack of strict invariance. All tests of invariance are part of a hierarchy, and these nested models are tested from the least restrictive (configural model) to the more restrictive test (strict invariance).

Model fit was evaluated and the presence or absence of ME/I was determined using the fit indexes and thresholds previously described. Additionally, the model χ^2^ is additive and allows for tests of the differences of chi-square values (Δχ^2^) between nested models to determine whether the tested level of invariance is accepted (i.e., is the inclusion of added restrictions, for example—constraining factor loadings to be equal across groups—acceptable or does it degrade the fit of the model?). The Δχ^2^ should be a non-significant difference for the test of invariance to be accepted. This indicates that the change from a less restrictive model to a more restrictive model is negligible [59]. One concern with this approach to comparing models is that Δχ^2^ values are overly sensitive. Additional methods for model comparison include computing differences or delta values for the CFI and RMSEA fit indexes (Table 1 and Table 2). The delta values for CFI would be rejected if the model change exceeds 0.005, or if delta for RMSEA exceeds 0.01 [63]. The process was similar for both physical activity/fitness and the HEI-C.

Modeling of associations in latent factors was then completed, to assess possible associations across the interventions.

## 3. Results

### 3.1. Physical Activity/Fitness

#### 3.1.1. Exploratory Factor Analysis Model

Pearson correlations were examined among the physical activity/fitness variables (i.e., treadmill speed, percentile VO_2_ max, partial curl-ups, push-ups, flexibility). Moderate to high correlations were observed for all but flexibility (range r = 0.36 between speed and push-ups to r = 0.85 between speed and VO_2_ max). Baseline reliability and test-retest correlations for the rest of the measures was good (α = 0.65; r_speed_ = 0.74, r_VO2max_ = 0.89, r_curlup_ = 0.66, r_pushup_ = 0.81). Flexibility was dropped from further consideration.

Exploratory Factor Analysis of the physical activity/fitness items, using Maximum Likelihood extraction demonstrated a one-factor model fit the data. The Kaiser–Meyer–Olkin (KMO) measure of sampling adequacy was only moderate with a value of 0.66. The KMO ranges from 0–1.00 with values greater than 0.8 providing evidence that the relationships among examined variables are amenable to factor analytic procedures. Individual items were assessed using the KMO values from the diagonal of the anti-image matrix. KMO values ranged from 0.61 for speed and VO_2_ max to 0.83 for partial curl-ups.

There was only one eigenvalue greater than 1.0 and the Scree Plot showed a sharp break between one and two factors, suggesting a one-factor solution. VO_2_ max is often used as a “gold-standard” measure of fitness, and our goal was to evaluate whether adding measures of strength would contribute to measuring fitness above and beyond cardiorespiratory measures. Model fit was moderate, with a model χ^2^ (2) = 24.12, *p* = 0.001. The Goodness of Fit Index (GFI) index in EFA should be non-significant, but one also has to interpret the factor loadings, which ranged from moderate to large (i.e., Λpush-ups = 0.44, Λcurl-ups = 0.46, Λspeed = 0.88, and ΛVO_2_ max = 0.97). The communalities (i.e., the amount of variance accounted for in each item) were 0.19 for push-ups, 0.21 for curl-ups, 0.77 for speed, and 0.93 for VO_2_ max. The overall sum of squared factor loadings or percentage of variance accounted for by the model was 53% utilizing the single-factor solution.

Upon confirmation of this result, we examined CFA utilizing the Analysis of Moments Structures (AMOS) Structural Equation Modeling (SEM) program (Amos Version 26.0. Chicago: IBM SPSS). The results from AMOS are presented below. All factor loadings were statistically significant, and the variance accounted for in the observed indicators varied between 18% and 97% (see Figure 2). These results map onto the EFA results described above. One benefit of the SEM/CFA approach is that AMOS provides numerous measures of model fit not available in EFA statistical packages. Overall model fit was strong and supported the multiple indicator model of fitness (see Figure 2 and Table 1, Models #2 and #3). The only modification of the factor model was correlating the error term between push-ups and curl-ups, as both were indicators of strength. 

**Table 1 nutrients-13-04258-t001:** Model Comparison for Physical Activity/Fitness Models.

Model #	Model	*X*^2^ (df)	*X*^2^ *p*-Value	CFI	NNFI	RMSEA [95% CI]	Δ*X*^2^ ΔCFI ΔRMSEA
	**Desirable Criterion or Range**		*NS desirable*	*>0.9*	*>0.9*	*0.05–0.08 acceptable;* *lower better*	Δ*X^2^* = *NS* Δ*CFI* ≤ *0.005* Δ*RMSEA* ≤ *0.01*
	**Longitudinal Invariance**						
1	1-Factor Model	0.939 (1)	0.333	1.00	1.00	0.000 0.000–0.153	
2	Longitudinal Configural	139.29 (37)	0.001	0.97	0.94	0.097 0.080–0.115	
3	*Longitudinal* *Metric*	*159.36* *(43)*	*0.001*	*0.96*	*0.93*	*0.096* *0.081–0.112*	*Reject.* *Accept* *Accept*
4	Longitudinal Intercepts Only	394.09 (45)	0.001	0.89	0.80	0.157 0.148–0.178	Reject Reject Reject
5	Longitudinal Loadings and Intercepts	415.82 (51)	0.001	0.88	0.82	0.157 0.143–0.171	Reject Reject Reject
6	Longitudinal Model Residuals	Not tested as invariant intercepts not found					
	**Sex Models—Female**						
7	Female Baseline	0.100 (1)	0.752	1.00	1.00	0.000 0.000–0.148	
8	Female Longitudinal Configural	56.99 (37)	0.019	0.98	0.96	0.060 0.025–0.089	
*9*	*Female* *Longitudinal* *Metric*	*66.57* *(43)*	*0.012*	*0.98*	*0.96*	*0.060* *0.029–0.088*	*Accept* *Accept* *Accept*
10	Female Intercepts Only	171.13 (45)	0.001	0.88	0.80	0.136 0.115–0.158	Reject Reject Reject
11	Female Loadings and Intercepts	180.89 (51)	0.001	0.88	0.82	0.130 0.110–0.151	Accept Accept Accept
12	Female Residuals	Not tested as invariant intercepts not found					
	**Sex Models—Male**						
13	Male Baseline	3.94 (1)	0.047	0.99	0.90	0.145 0.014–0.307	
14	Males Longitudinal Configural	117.59 (37)	0.001	0.95	0.90	0.125 0.100–0.150	
*15*	*Males* *Longitudinal* *Metric*	*136.59* *(43)*	*0.001*	*0.95*	*0.90*	*0.125* *0.102–0.149*	*Reject* *Accept* *Accept*
16	Male Intercepts Only	250.41 (45)	0.001	0.88	0.80	0.181 0.159–0.203	Reject Reject Reject
17	Male Loadings and Intercepts	275.94 (51)	0.001	0.87	0.80	0.177 0.157–0.198	Reject Reject Reject
18	Male Residuals	Not tested as invariant intercepts not found					
	**Gender Invariance of Longitudinal Fitness Model**						
19	Sex Invar. Configural	174.60 (74)	0.001	0.96	0.93	0.068 0.055–0.082	
*20*	*Sex Model* *Sex Invariant*	*180.76* *(83)*	*0.001*	*0.97*	*0.94*	*0.064* *0.051–0.076*	*Accept* *Accept* *Accept*
*21*	*Sex Model* *Time Invariant*	*206.58* *(89)*	*0.001*	*0.96*	*0.93*	*0.067* *0.055–0.079*	*Reject* *Accept* *Accept*
22	Sex Model Intercepts	Not run based on previous intercept models					
23	Sex Model Residuals	Not run as intercept models were not accepted					
	**Disease-State Models—No Diabetes**						
24	NoDM Baseline	1.12 (1)	0.290	1.00	0.96	0.029 0.000–0.228	
25	NoDM Longitudinal Configural	97.42 (37)	0.001	0.96	0.91	0.108 0.082–0.134	
*26*	*NoDM* *Longitudinal* *Metric*	*106.96* *(43)*	*0.001*	*0.95*	*0.91*	*0.103* *0.079–0.128*	*Accept* *Accept* *Accept*
27	NoDM Longitudinal Intercepts Only	218.98 (45)	0.001	0.87	0.78	0.166 0.145–0.189	Reject Reject Reject
28	NoDM Residuals	Not tested as invariant intercepts not found					
	**Disease-State Models—Diabetes**						
29	DM Baseline	0.002 (1)	0.968	1.00	1.00	0.000 0.000–0.000	
30	DM Longitudinal Configural	72.57 (37)	0.001	0.98	0.96	0.080 0.052–0.107	
*31*	*DM* *Longitudinal* *Metric*	*89.75* *(43)*	*0.001*	*0.97*	*0.95*	*0.085* *0.060–0.110*	*Reject* *Accept* *Accept*
32	DM Longitudinal Intercepts Only	215.14 (45)	0.001	0.91	0.84	0.158 0.137–0.180	Reject Reject Reject
33	DM Residuals	Not tested as invariant intercepts not found					
	**Disease Invariance of Longitudinal Fitness Model**						
34	Disease Model Configural	170.00 (74)	0.001	0.97	0.94	0.067 0.054–0.080	
*35*	*Disease Model* *Disease Invariant*	*184.87* *(83)*	*0.001*	*0.97*	*0.94*	*0.065* *0.052–0.078*	*Accept* *Accept* *Accept*
*36*	*Disease Model* *Time Invariant*	*201.93* *(89)*	*0.001*	*0.96*	*0.94*	*0.066* *0.054–0.078*	*Reject* *Accept* *Accept*
37	Disease Model Intercepts	Not run based on previous intercept models					
38	Disease Model Residuals	Not run as intercept models were not accepted					

*X^2^ =* Model chi square; CFI = Comparative Fit Index; NNFI = Non-Normed Fit Index; RMSEA = Root Mean Square Error of Approximation; Δ = change. Best model(s) in each hierarchal set of models shown in italics.

#### 3.1.2. Longitudinal Extension of Physical Activity/Fitness Model

Given the results above, we extended the measurement model in AMOS to test a longitudinal (i.e., three time points: baseline, 3-months, 12-months) model [59]. As we measured the same indicators of fitness on each occasion, the AMOS program and standard convention in SEM allows for correlations between the same indicators over time (e.g., speed at baseline is expected to correlate with the subsequent speed measures at 3- and 12-months). Each of the four indicators were treated in this manner, to allow for the autocorrelation of measuring the same indicators on the same participants, over time (see Figure 3). The extended, longitudinal model of fitness (Table 1, Model #2) demonstrated excellent model fit (χ^2^ (37) = 139.29, *p* < 0.001, CFI =.97, NNFI = 0.94, RMSEA= 0.097). All factor loadings were statistically significant, with squared multiple correlation or variance accounted for ranging from 8% to 87% (not shown).

The fitness factor created at each measurement point was regressed onto the subsequent measure (i.e., baseline fitness predicting 3-month fitness, 3-month fitness predicting 12-month fitness) and the results showed that 73% of the variance in fitness at 3-months was predicted by baseline fitness, and 84% of variance in fitness at 12-months was explained by fitness level at 3-months of the intervention (see Figure 3 and Table 1, Model #3). The model did not meet criteria for invariance of intercepts however, longitudinal invariance of factor loadings was found (Table 1, Models #4 and #5 compared to Model #3).

#### 3.1.3. Sex Invariance of Physical Activity/Fitness Model

Invariance testing was conducted for females (cross-sectional/baseline fitness model, longitudinal extension of fitness model (tests of invariance of variable loadings, intercepts, intercepts and loadings, and residuals if appropriate); males (testing the same sequence noted above); then examining the issue of sex invariance for the longitudinal fitness model (i.e., test of invariance of loading across time in a simultaneous model, constraining equality of the longitudinal loadings across sex, tests of equality of intercepts, and tests of residuals, if appropriate). Our results showed that, in addition to the model fitting well for both women (Model #9) and men (Model #15), invariance of the fitness model was demonstrated across the three time points of the intervention as well as across sex (Table 1, Models #20 and #21).

Testing equivalence of the variable intercepts were mixed at best, and based on overall decrement of model fit, it was deemed that invariant intercepts were not accepted (Table 1, Models #10 and #16). Given this result, and that all models are hierarchical regarding their restrictiveness (i.e., if one fails to accept equivalence of loadings, you should not continue with ME/I testing), we concluded that the fitness model demonstrated invariant factor loadings for sex and across time. The final result is shown in Figure 3. The magnitudes of the standardized loadings were identical for males and females, however, as we were unable to accept the constraint of the residual or uniqueness terms, the loadings appear slightly different in size over time (i.e., unstandardized loadings were identical across the three time points). Standardized loadings are presented as they are generally easier to interpret (with values ranging from 0–1, with larger values indicating stronger loadings) and they are similar to correlations for general interpretation purposes. VO_2_ max and speed measures contributed more to the physical activity/fitness factor than the strength measures (push-ups, curl-ups), yet all four measures are significant components of the overall factor.

#### 3.1.4. Disease-State Invariance of Physical Activity/Fitness Model

Invariance was tested for participants with noDM (cross-sectional/baseline fitness model, longitudinal extension of fitness model, tests of invariance of variable loadings, intercepts, intercepts and loadings, and residuals if appropriate); those with DM (testing the same sequence noted above); then examining the issue of disease-state invariance for the longitudinal fitness model (i.e., test of invariance of loadings across time in a simultaneous model, constraining equality of the longitudinal loadings across disease-state, tests of equality of intercepts, and tests of residuals, if appropriate). Results showed that, in addition to the model fitting well for participants with noDM (Table 1, Model #26), and those with DM (Model #31), invariance of the fitness model was demonstrated across the three time points of the intervention as well as across disease-states (Table 1, Models #35 and #36). Testing equivalence of the variable intercepts gave mixed results at best, and based on overall decrement of model fit, it was deemed that invariant intercepts were not accepted.

We found that the fitness model demonstrated invariant loadings for disease-states (noDM vs. DM) and across time (baseline, 3-months, 12-months). The results for the disease-state model are shown in Figure 4. The magnitude of the standardized loadings are identical for noDM and DM, however, as we were unable to accept the constraint of the residual or uniqueness terms, the loadings appear slightly different in size over time (i.e., unstandardized loadings were identical across the three time points and across disease-states). As noted in the previous results, VO_2_ max and speed measures contributed more to the physical activity/fitness factor than the strength measures (push-ups, curl-ups), yet all four measures are significant components of the overall factor. It was noted that the amount of explained variance in the physical activity/fitness factor was higher in the DM group (73% at 3-months, 91% at 12-months, than the noDM group: 73% at 3-months, 75% at 12-months).

### 3.2. Healthy Eating Index (HEI-C)

#### 3.2.1. Exploratory Factor Analysis Model

Pearson correlations were examined for the 11 HEI-C items (see Supplement, Appendix A). Low to moderate correlations were observed, with some items showing very small correlations (e.g., milk and alternatives, unsaturated fats, total grains, and meat and alternatives showed the lowest correlations). We examined EFA of the full complement of 11 HEI-C items, using Maximum Likelihood extraction, with Promax rotation if the result had more than one resulting factor (i.e., to allow the resulting factors to correlate). The Kaiser-Meyer-Olkin (KMO) measure of sampling adequacy was only moderate with a value of 0.604. Individual items were assessed using the KMO values from the diagonal of the anti-image matrix. Variables with the lowest KMO values (i.e., below 0.6) were: total grains, meat and alternatives, milk and alternatives, raising concern about the inclusion of these items.

There were four eigenvalues exceeding 1.0, however, the Scree Plot showed a gradual decline without any clear breaks or drop-offs, suggesting a one-factor solution was most likely. Four factors with only 11 items would not be reasonable (i.e., ideally three or more items should result per factor), and the one-factor solution, especially if weak items were going to be removed/deleted, made conceptual sense. We evaluated the four-factor solution (based on eigenvalues greater than 1.0). Model fit was poor, with a model χ^2^ (17) = 26.59, *p* = 0.064 and factor loadings were not conceptually meaningful.

The total score of the HEI-C is most widely used, therefore a one-factor solution would provide evidence whether this is a valid and meaningful approach. We examined a one-factor HEI-C model. The resulting factor loadings for the 11-item HEI-C EFA still reflected the same items noted earlier as being weak in this solution (i.e., milk and alternatives, meat and alternatives, total grains, and unsaturated fats). While many of these items make conceptual sense or might be seen as useful in dietary models, statistical evidence was not supporting retention of these items. We also noted that the communalities (i.e., the amount of variance accounted for in each item) were very low for these four items.

Additional EFA with 7 HEI-C items (i.e., removing the poorly fitting items: milk and alternatives, meat and alternatives, total grains, and unsaturated fats), Maximum Likelihood extraction, and no rotation in one factor was examined. The Kaiser–Meyer–Olkin (KMO) measure of sampling adequacy improved to 0.675 but was still low. Individual items ranged from 0.621–0.780.

There were two eigenvalues exceeding 1.0 (the second values barely exceeded 1.0 at 1.09), the Scree Plot showed a gradual decline without any clear breaks or drop-offs, suggesting a one-factor solution. Model fit was improved from the previous model, but it is still not an ideal model χ^2^ (14) = 35.97, *p* = 0.001. Factor loadings ranged from 0.25–0.85.

#### 3.2.2. Testing the Reduced HEI-C in CFA/SEM

The results of testing the reduced HEI-C model in a single factor solution with seven items in AMOS are presented in Figure 5. All factor loadings were statistically significant (low of 0.34 for whole Grains to a high of 0.50 for total vegetables/Fruit) and the variance accounted for ranged between 12% (whole grains) to 25% (vegetables/fruit), with an average variance of 18%. These results are consistent with the EFA results described above. We added two correlated error/residual terms to the model. Total vegetables/fruits and whole fruits, and also between total vegetables/fruits and dark green and orange vegetables. These variables are conceptually related. Model fit for the 7-item version (Table 2, Model #1) was exceptional with χ^2^ (12) = 11.10, *p* < 0.521, CFI = 1.00, and RMSEA = 0.000. For comparison purposes, the 11-item (full) HEI-C resulted in two items with non-significant factor loadings (i.e., total grains and meat and alternatives), and both milk and alternatives and unsaturated fats had marginal/borderline values and χ^2^ (44) = 208.28, *p* < 0.001, CFI= 0.57, and RMSEA = 0.11, outside the published cut-off values range (0.05 to 0.08).

The reduced-item HEI-C model was therefore tested for longitudinal invariance in the total sample, as well as for each sex and each disease group (i.e., noDM/DM). The model demonstrated weak metric invariance (i.e., equivalence of the magnitude of the factor loadings over all three measurement points in the intervention—Table 2, Model #3). The model also demonstrated weak invariance of all factor loadings when compared across sex (Model #20) and across disease groups (Model #35).

**Table 2 nutrients-13-04258-t002:** Model Comparison for Reduced (7-Item) HEI-C Models.

Model #	Model	*X*^2^ (df)	*X*^2^ *p*-value	CFI	NNFI	RMSEA [95% CI]	Δ*X*^2^ ΔCFI ΔRMSEA
	**Desirable Criterion or Range**		*NS desirable*	*>0.9*	*>0.9*	*0.05–0.08 acceptable;* *lower better*	Δ*X2 = NS* Δ*CFI ≤ 0.005* Δ*RMSEA ≤ 0.01*
	**Longitudinal Invariance**						
1	1-Factor Model	11.10 (12)	0.521	1.00	1.00	0.000 0.000 -0.056	
2	Longitudinal Configural	205.15 (160)	0.009	0.95	0.93	0.031 0.046–0.063	
*3*	*Longitudinal* *Metric*	*219.33* *(172)*	*0.009*	*0.95*	*0.94*	*0.031* *0.016–0.042*	*Accept* *Accept* *Accept*
4	Longitudinal Intercepts Only	371.10 (174)	0.001	0.80	0.74	0.062 0.054–0.071	Reject Reject Reject
5	Longitudinal Loadings and Intercepts	388.17 (186)	0.001	0.80	0.75	0.061 0.052–0.070	Accept Accept Accept
6	Longitudinal Model Residuals	Not tested as invariant intercepts not found					
	**Sex Models—Female**						
7	Female Baseline	25.39 (12)	0.019	0.90	0.77	0.086 0.038–0.133	
8	Female Longitudinal Configural	200.85 (160)	0.016	0.93	0.89	0.041 0.019–0.058	
*9*	*Female * *Longitudinal* *Metric*	*219.28* *(172)*	*0.009*	*0.92*	*0.89*	*0.043* *0.028–0.059*	*Accept* *Accept* *Accept*
10	Female Intercepts Only	290.84 (174)	0.001	0.79	0.72	0.067 0.053–0.080	Reject Reject Reject
11	Female Loadings and Intercepts	309.48 (186)	0.001	0.78	0.72	0.066 0.053–0.079	Accept Accept Accept
12	Female Residuals	Not tested as invariant intercepts not found					
	**Sex Models—Male**						
13	Male Baseline	8.11 (12)	0.777	10.00	10.00	0.000 0.000–0.059	
14	Males Longitudinal Configural	210.31 (160)	0.005	0.90	0.85	0.047 0.027–0.064	
*15*	*Males* *Longitudinal* *Metric*	*219.33* *(172)*	*0.009*	*0.95*	*0.94*	*0.031* *0.016–0.042*	*Accept* *Accept* *Accept*
16	Male Intercept Only	310.29 (174)	0.001	0.72	0.63	0.072 0.061–0.088	Reject Reject Reject
17	Male Loadings and Intercepts	324.04 (186)	0.001	0.72	0.65	0.073 0.059–0.086	Reject Reject Reject
18	Male Residuals	Not tested as invariant intercepts not found					
	**Sex Invariance of Longitudinal HEI-C Model**						
19	Sex Invar. Configural	415.75 (320)	0.001	0.94	0.91	0.026 0.018–0.033	
*20*	*Sex Model* *Sex Invariant*	*420.65* *(338)*	*0.001*	*0.92*	*0.88*	*0.030* *0.019–0.038*	*Accept* *Accept* *Accept*
*21*	*Sex Model* *Time Invariant*	*454.23* *(350)*	*0.001*	*0.90*	*0.87*	*0.032* *0.023–0.040*	*Reject* *Accept* *Accept*
22	Sex Model Intercepts	Not run based on previous intercept models					
23	Sex Model Residuals	Not run as intercept models were not accepted					
	**Disease-State Models—No Diabetes**						
24	NoDM Baseline	16.12 (12)	0.186	0.96	0.91	0.059 0.000–0.106	
25	NoDM Longitudinal Configural	205.10 (160)	0.009	0.91	0.87	0.045 0.023–0.062	
*26*	*NoDM* *Longitudinal* *Metric*	*224.15* *(172)*	*0.005*	*0.89*	*0.86*	*0.047* *0.027–0.063*	*Accept* *Accept* *Accept*
27	NoDM Longitudinal Intercepts Only	283.75 (174)	0.001	0.77	0.70	0.067 0.053–0.081	Reject Reject Reject
28	NoDM Residuals	Not tested as invariant intercepts not found					
	**Disease-State Models—Diabetes**						
29	DM Baseline	9.50 (12)	0.660	1.00	1.00	0.000 0.000–0.068	
30	DM Longitudinal Configural	172.59 (160)	0.235	0.98	0.96	0.023 0.000–0.045	
*31*	*DM* *Longitudinal* *Metric*	*183.88* *(172)*	*0.254*	*0.98*	*0.97*	*0.021* *0.000–0.043*	*Accept* *Accept* *Accept*
32	DM Longitudinal Intercepts Only	276.63 (174)	0.001	0.80	0.73	0.063 0.048–0.076	Reject Reject Reject
33	DM Residuals	Not tested as invariant intercepts not found					
	**Disease Invariance of Longitudinal HEI-C Model**						
34	Disease Model Configural	377.70 (320)	0.015	0.95	0.92	0.025 0.012–0.034	
*35*	*Disease Model* *Disease Invariant*	*397.95* *(338)*	*0.014*	*0.94*	*0.92*	*0.025* *0.012–0.034*	*Accept* *Accept* *Accept*
*36*	*Disease Model* *Time Invariant*	*410.90* *(350)*	*0.014*	*0.94*	*0.92*	*0.024* *0.012–0.034*	*Accept* *Accept* *Accept*
37	Disease Model Intercepts	Not run based on previous intercept models					
38	Disease Model Residuals	Not run as intercept models were not accepted					

*X^2^ =* Model chi square; CFI = Comparative Fit Index; NNFI = Non-Normed Fit Index; RMSEA = Root Mean Square Error of Approximation; Δ = change. Best model(s) in each hierarchal set of models shown in italics. This explains why the acceptable model is the longitudinal metric models (Model #3 and #9) and not the loadings and intercepts models (Model #5 and #11).

#### 3.2.3. Longitudinal Extension of Reduced HEI-C Model

Based on the cross-sectional results above, we extended the reduced (7-item) HEI-C model to test the fit longitudinally. As with the physical activity/fitness models previously, we allowed correlations between the same indicators over time (e.g., vegetable/fruit at baseline are expected to correlate with the subsequent vegetable/fruit measures). We also maintained the correlation between the vegetable/fruit variable error terms as described above. The autocorrelation of the same indicators on the same participants over time was evaluated (see Appendix A). The extended, longitudinal model demonstrated excellent model fit (Table 2, Model #3). All measures of model fit improved for the longitudinal HEI-C model compared to the cross-sectional or baseline model. All factor loadings were statistically significant with the exception of the sodium variable at 3-months, and squared multiple correlations or variance accounted for in each variable, ranging from just over 1% (sodium at 3-months) to 72% for vegetables/fruit at baseline.

The HEI-C factor was regressed onto the subsequent measure (i.e., baseline HEI-C predicting 3-month HEI-C, 3-month HEI-C predicting 12-month HEI-C) and the results showed that 42% of the variance in HEI-C at 3-months was predicted by baseline HEI, and 52% of variance in HEI-C at 12-months was explained by HEI-C at 3-months of the intervention (see Appendix A).

#### 3.2.4. Sex Invariance of Reduced HEI-C Model

Invariance testing followed the same sequence as for the physical activity/fitness model. Models were examined for females (cross-sectional/baseline HEI-C model, longitudinal extension of HEI-C model (tests of invariance of variable loadings, intercepts, intercepts and loadings, and residuals if appropriate); males (testing the same sequence noted above); then examining the issue of sex invariance for the longitudinal HEI-C model (i.e., test of invariance of loading across time in a simultaneous model, constraining equality of the longitudinal loadings across sex, tests of equality of intercepts, and tests of residuals, if appropriate). Results showed that, in addition to the model fitting well for both men (Model #15) and women (Model #9), invariance of the HEI-C model was demonstrated across the three time points of the intervention as well as across sex (Models #20 and #21). Testing equivalence of the variable intercepts yielded results that were mixed at best, and based on overall decrement of model fit, it was deemed that invariant intercepts were not accepted. The HEI-C model demonstrated invariant loadings for sex and across time. The final result is shown in Appendix A.

The magnitudes of the standardized loadings are identical for males and females, however, as we were unable to accept the constraint of the residual or uniqueness terms, the loadings appear slightly different in size over time (i.e., unstandardized loadings were identical across the three time points).

#### 3.2.5. Disease State Invariance of Reduced HEI-C Model

Invariance testing participants with and without DM (cross-sectional/baseline HEI-C model, longitudinal extension of HEI-C model, tests of invariance of variable loadings, intercepts, intercepts and loadings, and residuals if appropriate); then examining the issue of disease-state invariance for the longitudinal HEI-C model (i.e., test of invariance of loading across time in a simultaneous model, constraining equality of the longitudinal loadings across disease-state, tests of equality of intercepts, and tests of residuals, if appropriate). Results showed that, in addition to the model fitting well for participants with no DM (Model #26), and those with DM (Model #31), invariance of the HEI-C model was demonstrated across the three time points of the intervention as well as across disease-states (Models #35 and #36). Testing equivalence of the variable intercepts yielded poor results, and invariant intercepts were not accepted. We found that the HEI-C model demonstrated invariant loadings for disease-states (noDM vs DM) and across time (Appendix A).

### 3.3. Assessment of Associations between Physical Activity/Fitness and Reduced HEI-C

To test whether diet quality and physical activity/fitness were significantly related over the course of the year-long intervention, structural regression was conducted. For males and females overall, regression paths in green showed significant results, structural regression in red were not statistically significant (Figure 6). The following relationships were significant: fitness-baseline to HEI-C-3month; HEI-C-3-month to fitness-12-month; fitness-3-month to HEI-C-12-months.

The models were then examined for sex and disease-state comparisons. Previous results suggested that both models were at least weak-metric invariant (i.e., the factor loadings for both physical activity/fitness and HEI-C, demonstrated equivalence across the comparison group and over time). This result was found, with the sex-model demonstrating weak-metric invariance as the highest level χ^2^ (935) = 1375.33, CFI = 0.89, NNFI = 0.87, RMSEA= 0.040. Strong auto-regressive relationships were noted for fitness, wherein fitness-baseline predicted 53% of the variance of fitness-3-month, and fitness-3-month predicted 64% of the variance in fitness at 12-months in females, whereas these values were 64% and 84% respectively in males. Regarding HEI-C, baseline to 3-months accounted for 39% variance and 3-month to 12-months accounted for 57% in females, whereas 51% and 40% variance were accounted for in males.

We then examined the relationships separately for women and men (Appendix A). For women, the only significant relationship was fitness at baseline was significantly related to HEI-C at 3-months (Appendix A), while for men HEI-C at 3-months was significantly related to fitness at 12-months and fitness at 3-months predicted HEI-C at 12-months (Appendix A).

The disease-state model showed invariance through the level of structural variances/covariance (i.e., the only coefficients in the model that were not equivalent were the residual or error terms). Strong auto-regressive relationships were noted for fitness whereby fitness-baseline predicted 70% of the variance of fitness-3-month, and fitness-3-month predicted 91% of the variance in fitness at 12-months in DM, whereas these values were 74% and 78% respectively in participants with noDM. Regarding HEI-C, baseline to 3-months accounted for 44% variance and 3-month to 12-months accounted for 41% in DM, whereas 42% and 67% variance were accounted for in noDM. Model fit: χ^2^ (978) = 1261.94, CFI = 0.93, NNFI = 0.92, RMSEA= 0.032. Of note among the structural regressions, only the relationship from HEI-C-3-month to fitness-12-month was statistically significant in participants with noDM and those with DM (see Appendix A).

## 4. Discussion

Health behavior change researchers working in community and primary care contexts are interested in the potential for using composite summary scales and measures to describe changes in health behaviors. Scales have typically been developed and validated by researchers within the nutrition and kinesiology disciplines and it is common to adopt validated tools in community intervention studies, as was done in this secondary analysis. To answer our original question, could measurement error have accounted for the lack of association with the CVD risk score? Certainly, the measurement properties of the original HEI-C were poor and could have contributed to a lack of association. Lack of association of changes in percentile VO_2_ max could also be due to measurement issues with VO_2_ max, especially given recent documentation of measurement error in similar equations tested by Peterman et al. [33]. The Ebbeling equation was not specifically tested in their validation study on multiple equations against measured VO_2_ max. Further work on the measurement properties of both diet quality and fitness measures is warranted.

The results of the analysis of the measurement properties of HEI-C were particularly interesting as the concept of using scales to assess overall diet quality has a relatively long history in nutritional epidemiology, with the first HEI published in 1995, based on the work of Kennedy and colleagues [64]. Four measures within the original 11-item HEI-C model did not contribute to the latent HEI-C factor: milk and alternatives, meat and alternatives, total grains, and unsaturated fats. While these food groups are important for general health, if intake did not vary among the participants with low vs. high HEI scores at baseline and did not change with intervention, they will not contribute to the model. Examination of the data from the original study provides indirect evidence to support this interpretation [50]. Mean HEI scores for meat and alternates were high at baseline (8/10 possible points) and remained high throughout the 12-month intervention. Scores for milk and alternates were average at baseline (4.8 of 10 possible points), increased slightly at 3-months and returned to 5.0 at 12-months, reflecting no change attributable to the dietary intervention. Scores for total grains declined from 3.4 of 5 possible points to 3.1 at 12-months, while carbohydrate intake was 48% of kcal throughout. Unsaturated fat intake was confirmed to be low and relatively stable both from the nutrient analysis and HEI-C analysis. The reduced latent HEI-C factor had better measurement properties and it was possible to detect interesting interactions between physical activity and diet change.

Going forward, development of new diet quality tools is needed for the intervention context. Such intervention tools can potentially be adapted from tools already developed in epidemiology. Interesting work is underway to develop an adapted diet quality tool for Canada that is associated with lower risk of MetS markers. Lafreniere et al. have used reduced rank regression to identify food groups associated with MetS markers in a French Canadian sample [65]. Their retained food groups in their modified C-HEI tool were total vegetables and fruit, whole fruit, dark green and orange vegetables, whole grains, yogurt, nuts and legumes, red and processed meat, refined grains, sugar-sweetened beverages, “other foods”. While there is substantial overlap with our reduced 7-item HEI-C, namely five of seven items are on both lists (total vegetables and fruit, whole fruit, dark green and orange vegetables, whole grains “other foods”) with only saturated fat and sodium retained in our version but not in their version. Interestingly, both our work and Lafreniere et al. use variance as a key criterion for retention of food groups. There is some danger to this data-driven approach, in that food groups that are important to health may be missed. For example, fish may be important; it is a key component of the Mediterranean diet with desirable nutritional properties, yet Canadians do not eat very much fish and it did not emerge as a group in the LaFreniere analysis. For future intervention studies, fish might need to be included as a target food to increase intake. Still, Lafreniere’s results are very interesting, and along with our work provide a basis for progress in the development of new diet quality tools for intervention work.

Parallel work in physical activity/fitness is needed but was not reviewed in detail. A one factor model that included cardiorespiratory fitness, treadmill speed, and measures of strength was generalizable across comparison groups, providing stable, reliable measurement of true mean differences in the latent fitness factor. The results confirm the relevance of VO_2_ max as a prominent contributor and that muscular strength is also relevant. The flexibility measure did not contribute, yet it may be possible to find other indicators of flexibility which could contribute to a summary measure. The results contribute to ongoing development and validation of summary physical activity/fitness scales suitable for community studies.

The results of the analysis showing that diet quality and physical activity/fitness were associated over time was informative, especially in that the effect was more prominent in men than women, with change in physical activity/fitness more likely to affect later changes in HEI-C, rather than diet changes influencing fitness later in the intervention. No comparable analyses were found in the literature. Recent studies in multiple health behavior change have developed composite summary scores of health behavior change [66], used other analysis methods [67] or focused on associations with other measures such as self-efficacy [68]. These preliminary results do suggest different strategies for programming with men and women and tend to support longer term interventions.

SEM has been used extensively in psychology, given the need to analyze latent constructs and non-independent longitudinal data within and across individuals and use is gradually increasing in nutrition and kinesiology, with the increasing interest in multi-dimensional scales for assessing diet quality and physical fitness change. Expertise is required to do the analysis; hence the process was more thoroughly explained than is typical Most past validation studies in diet and physical fitness have sought to establish means and ranges to be similar to some more accurate standard, with comparison to specific nutrients or physiologic measures, a basic strategy. However, if the other measurement properties (i.e., variances and covariances, variable intercepts, and residuals) addressed by SEM are not invariant, then the likelihood of detecting measurable change decreases dramatically. Further analyses, including structural regressions between latent constructs or development of latent change models, may be conducted with confidence that measurement error has been minimized and observed outcomes or differences are true differences and not driven by inaccurate measurement or imprecise methodology.

SEM is not without limitations [69,70]. There are methodological challenges in dealing with non-normal and missing data, both common in lifestyle intervention studies. Modelling complex phenomena is inherently challenging and use of SEM does not solve the inherent problem of model mis-specification and omitted variables. In addition, study design features such as the number of time points, number of indicators and their reliability will influence the power of SEM analysis. Generally, sample sizes in the range of ~200 are needed. Tomarken and Waller provide a useful introduction to strengths and limitations [70].

Expertise in both the content area and SEM is required to do the analysis; hence the analytical process was more thoroughly explained than is typical. Some invariance papers will not include all tests, however, the sequence should be examined in the appropriate order (i.e., some studies do not test intercept invariance but will test configural, weak, and strict invariance or others will include intercept invariance and stop without testing strict invariance). When the result of an invariance test is rejected or implausible, testing stops (e.g., if weak metric invariance was accepted/plausible, but invariance of intercepts rejected, we revert to the weak metric model as the accepted level for that model). Additional tests of structural (not measurement) invariance exist, when relationships between latent factors are tested for equality across groups/time (i.e., constraining factor variances/covariances or structural regression coefficients between factors). There are often questions of interest in these structural relationships of latent variables, but they are outside the realm of measurement invariance [71,72]. Regardless of the labels applied, demonstration of invariance is critically important in establishing the validity and application of the resulting factor models. Collaborators with expertise in SEM should be brought into the validation process much earlier and could be informing initial development of new diet and physical fitness measures for community intervention studies.

In conclusion, assessment of the measurement properties of diet quality and physical fitness measures was the main focus of this analysis, with an example analysis of possible associations between different lifestyle interventions. The original issue that prompted exploration of SEM; lack of association of intervention measures with disease risk score can now be partially explained. Work is underway to address the “mechanisms of action” for lifestyle programs in CMR conditions and we look forward to meaningful progress to improve effectiveness of lifestyle programs in health care and community settings.

## Figures and Tables

**Figure 1 nutrients-13-04258-f001:**
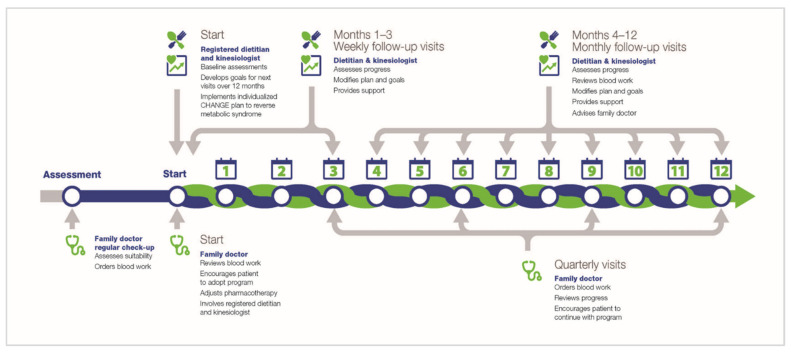
Data collection plan for lifestyle intervention study.

**Figure 2 nutrients-13-04258-f002:**
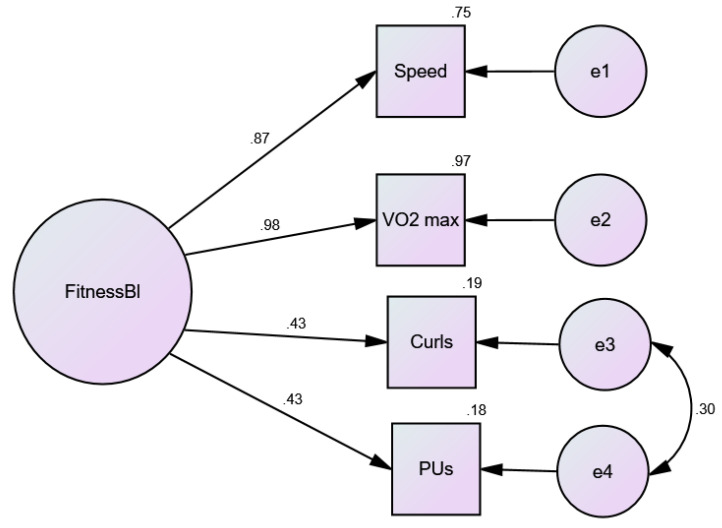
Baseline confirmatory factor analysis model. FitnessBl = latent factor; Speed = treadmill speed; VO_2_ max = age-sex percentile of VO_2_ max; Curls = Curl-ups; Pus = Push-ups; e# = error terms. Squares are measured variables; circles are latent variables.

**Figure 3 nutrients-13-04258-f003:**
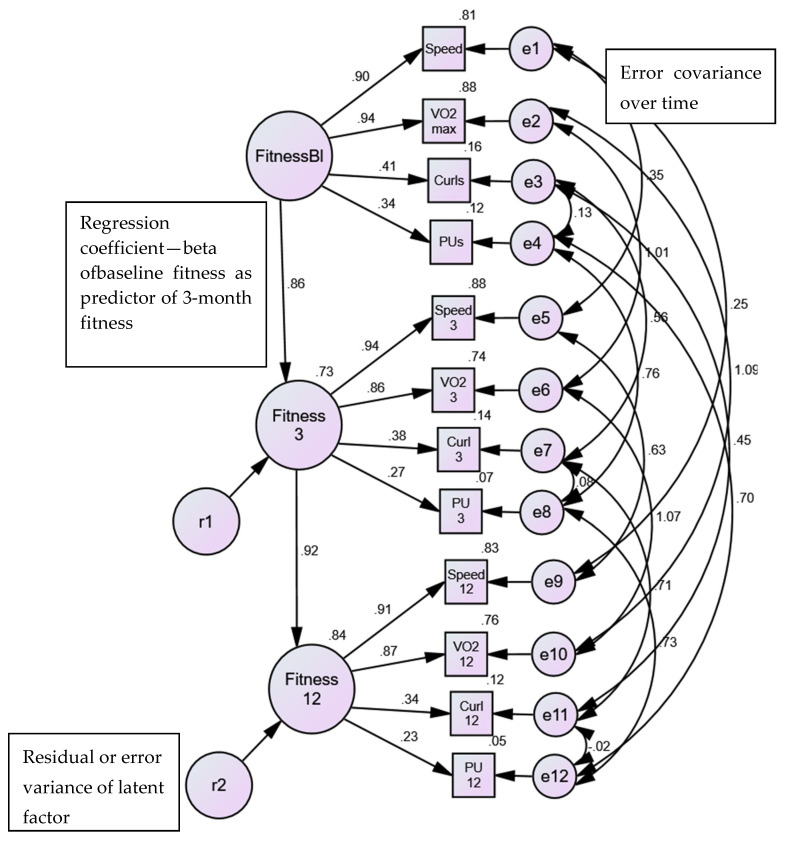
Sex and time invariance. FitnessB1 = baseline; Fitness3 = 3-months; Fitness12 = 12-months Speed = treadmill speed; VO_2_ max = age-sex percentile; Curls = Curl-ups; PU = Push-ups; e# and r# = error terms. Squares are measured variables; circles are latent variables.

**Figure 4 nutrients-13-04258-f004:**
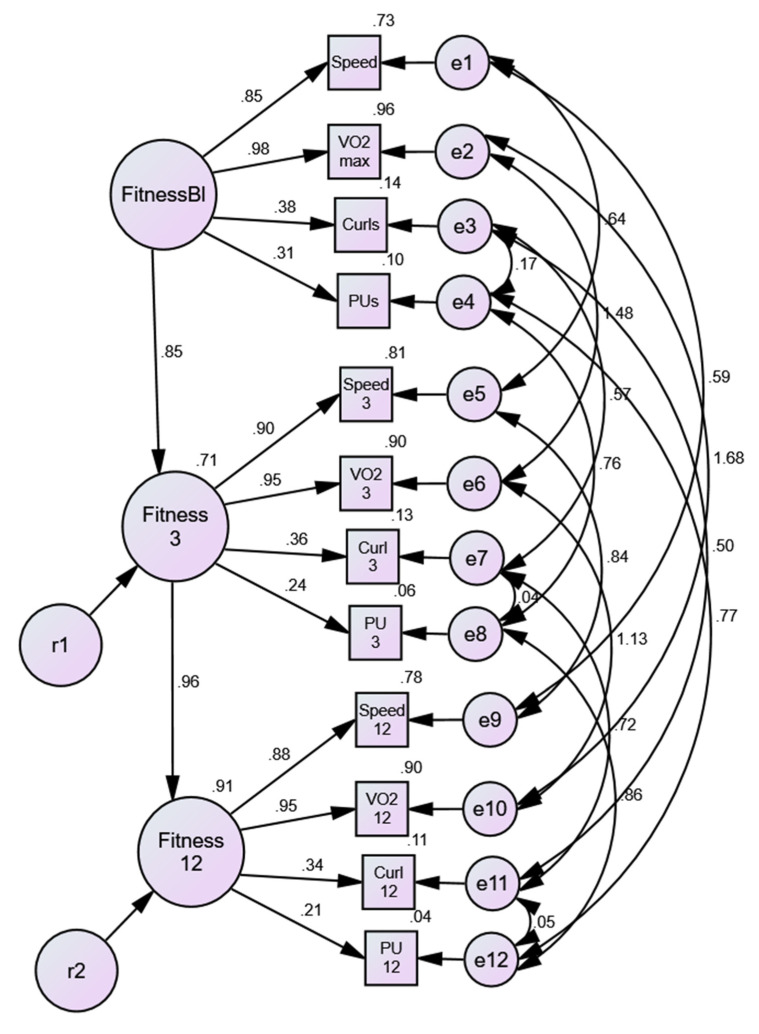
Disease invariance model. FitnessB1 = baseline; Fitness3 = 3-months; Fitness12 = 12-months Speed = treadmill speed; VO_2_ max = age-sex percentile; Curls= Curl-ups; PU= Push-ups; e# and r# = error terms.

**Figure 5 nutrients-13-04258-f005:**
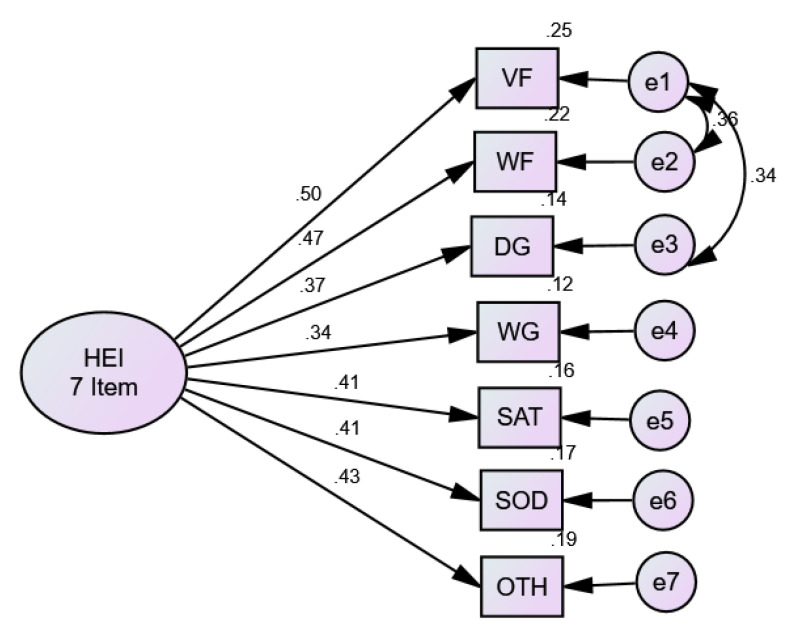
Baseline confirmatory factor analysis model for HEI-C. HEI = total HEI-C; VF = total vegetables and fruit; WF = whole fruit; DG = dark green and orange vegetables; WG = whole grains; SF = saturated fats; SOD = sodium; OTH = Other foods; e# = error terms. Squares are measured variables; circles are latent variables.

**Figure 6 nutrients-13-04258-f006:**
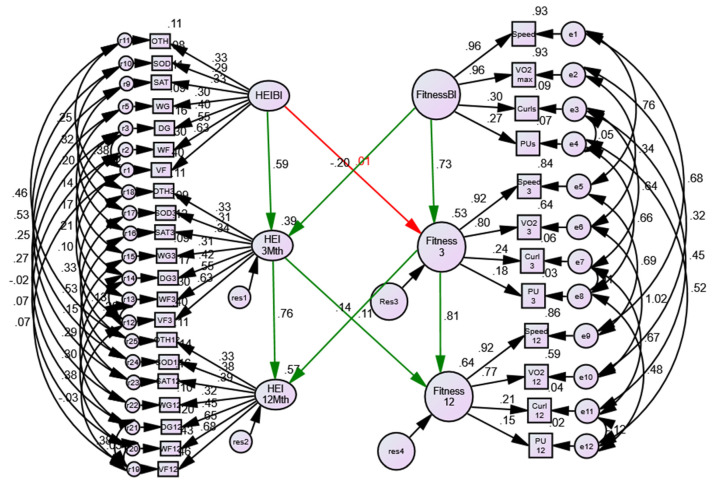
Overall sex and time invariant results showing structural regressions between HEI and Fitness factors. Regression paths in green show significant results, structural regression in red were not statistically significant. FitnessB1 = baseline; Fitness3 = 3-months; Fitness12 = 12-months Speed = treadmill speed; VO_2_ max = age-sex percentile; Curls = Curl-ups; PU = Push-ups. HEIBl = total HEI-C baseline; HEI3Mth = total HEI-C at 3-months; HEI12Mth = total HEI-C at 12-months; VF = total vegetables and fruit; WF = whole fruit; DG = dark green and orange vegetables; WG = whole grains; SF = saturated fats; SOD = sodium; OTH = Other foods; e# and r# = error terms.

## Data Availability

The datasets used and/or analysed during the current study are available from the corresponding author on reasonable request.

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
