# Peer review of "Evaluation of Latent Models Assessing Physical Fitness and the Healthy Eating Index in Community Studies: Time-, Sex-, and Diabetes-Status Invariance"

_nutrients, 2021, doi:10.3390/nu13124258_

Round 1

Reviewer 1 Report

Dear authors:

thanks for submitting this interesting article to Nutrients.

You made good and precise introduction. The 3 measurement issues you raised in your paper are relevant to research in promoting healthy lifestyle. The longitudinal desgin of your study is adequate, and fitness and diet assessments analysis are timely. The methodology used is correct and well executed. 

Generally speaking, some issues should be resolved in relation to the representation of the figures 2, 3 and especially 6, where the measured variables should be aligned in order to present more accurate models. The solution should be aligning properly the measured variables and adjust the abreviations of these variables into the squares.

Moreover I would suggest to have a separate limitations section at the end of the discussion section.

Thanks again for this necessary work.

Author Response

Regarding the presentation of the figures, we have redone some figures to improve presentation. Figure 2 has been adjusted so that variable abbreviations fit into the squares.  Figures 3, 4 and 6 variable abbreviations have been adjusted to fit within squares.  Supplement figures for HEI analysis have not been adjusted as they are understandable as is. 

Regarding discussion of limitations, we have adjusted paragraphs on limitations at lines 658- 681 and added 2 additional references on limitations of SEM.  We did not add a separate subtitle on Limitations.

Reviewer 2 Report

The work presented is very detailed and difficult to understand for the reader. It would be preferable to present the data in a more explanatory way. Personally in the statistical analysis I got lost and I see a great confusion.

Author Response

As noted by Reviewer 1, there is a need for more training and use of SEM in lifestyle interventions methods work.  We are sorry reviewer 2 did not find the study understandable.  The paper has been a collaboration of content and methods experts, precisely because a “worked” example can be helpful in prompting more learning by lifestyle intervention researchers.  We have not changed the manuscript on the basis of this review.     

Round 2

Reviewer 2 Report

the manuscript can be accepted in this last form

Author Response

thanks!